# The computational and neural substrates of moral strategies in social decision-making

Jeroen M. van Baar[1,2], Luke J. Chang [3] & Alan G. Sanfey[1,4]

Individuals employ different moral principles to guide their social decision-making, thus expressing a specific 'moral strategy'. Which computations characterize different moral strategies, and how might they be instantiated in the brain? Here, we tackle these questions in the context of decisions about reciprocity using a modified Trust Game. We show that different participants spontaneously and consistently employ different moral strategies. By mapping an integrative computational model of reciprocity decisions onto brain activity using inter-subject representational similarity analysis of fMRI data, we find markedly different neural substrates for the strategies of 'guilt aversion' and 'inequity aversion', even under conditions where the two strategies produce the same choices. We also identify a new strategy, 'moral opportunism', in which participants adaptively switch between guilt and inequity aversion, with a corresponding switch observed in their neural activation patterns. These findings provide a valuable view into understanding how different individuals may utilize different moral principles.

[1] Donders Institute for Brain, Cognition and Behavior, Radboud University, Nijmegen 6525 EN, The Netherlands. [2] Department of Cognitive, Linguistic, and Psychological Sciences, Brown University, Providence, RI 02912, USA. [3] Department of Psychological and Brain Sciences, Dartmouth College, Hanover, NH 03755, USA. [4] Behavioral Science Institute, Radboud University, Nijmegen 6525 HR, The Netherlands. Correspondence and requests for materials should be addressed to J.M.V.B. (email: jeroen_van_baar@brown.edu)

It is a well-worn moral adage that you should treat others as you yourself would like to be treated (the "golden rule"). Heated political debates on issues like immigration and health care, however, demonstrate that one can easily infuriate others by treating them according to one's own moral views. Often, the underlying theme of such debates is not policy, but rather the principle by which moral decisions are made.[1,2] For example, should we prioritize the principle of property or of equity, of solidarity or freedom? In diverse societies, different individuals may employ different sets of such fundamental priorities, thus expressing different "moral strategies". Such a strategy likely shapes not just political decisions but also behavior in everyday social interactions. In the present study, we sought to computationally characterize several distinct moral strategies in the context of reciprocity decisions, and map how these strategies are instantiated in the human brain.

Multiple moral motives have been proposed to explain reciprocity behavior, including preferences for consequentialism[3], in which people seek fairness in outcomes (inequity aversion[4,5]), and sentimentalism[6], in which people are motivated by feelings such as guilt in order to avoid harming others (guilt aversion[7])[8]. While previous neuroscientific investigations of these motivations have identified candidate brain regions involved in their computation, such as the anterior insula (AI), dorsolateral prefrontal cortex (DLPFC), anterior cingulate cortex (ACC), and ventromedial prefrontal cortex (VMPFC) for inequity aversion[9–12], and AI, VMPFC, DLPFC, supplementary motor area (SMA), and temporoparietal junction (TPJ) for guilt aversion[13,14], several important questions remain open. First, in most laboratory paradigms, guilt aversion and inequity aversion yield the same behavioral predictions, obfuscating which prosocial motivation was at play in the decision-maker's mind (as noted by Nihonsugi et al.[14] and Hein et al.[15]). Therefore, one important outstanding question is whether these two motivations can be uncoupled behaviorally. Second, the stability of moral strategies is largely unknown. Do people behave consistently across different instances of moral dilemmas or is their decision-making a product of the particular context they are facing? Finally, and crucially, previous neuroimaging studies have averaged measurements across participants, potentially masking individual differences in implicit moral reasoning, and hence obscuring strategy-specific features of the moral brain. Can we identify brain representations involved in processing computations specific to guilt aversion and inequity aversion? Can these be used in order to gain deeper insights into the nature of social decision-making?

To address these questions, we designed the Hidden Multiplier Trust Game (HMTG; Fig. 1), which can elicit behavioral differences in the decision to reciprocate trust as a function of an individual's moral strategy. In addition, we developed a computational model to identify distinct moral strategies, including behavioral patterns that reflect a shift in strategy across contexts. Finally, we sought to identify brain processes associated with different moral strategies using methods that leverage endogenous variation across participants.

Fifty-seven participants played the HMTG while undergoing functional magnetic resonance imaging (fMRI). On each trial of the HMTG, an anonymous Investor can send any number of 10 game tokens to the Trustee (the participant in the scanner), while retaining the remainder. As in traditional Trust Games[16], the Investor believes his investment will be multiplied by a fixed factor by the experimenter (here ×4) before being transferred to the Trustee. However, in the HMTG, only the Trustee knows that the actual multiplier alternates between ×6 (25% of trials), ×4 (50% of trials), and ×2 (25% of trials). Crucially, the Trustee is aware of the Investor's ignorance as to the actual multiplier, and knows that the Investor believes the multiplier is ×4 on

every trial. Therefore, on 25% of trials (the ×6 multiplier) the Trustee has more tokens than the Investor believes, and on 25% of trials (×2) they possess fewer tokens than the Investor thinks. Following the transfer, the Trustee now can choose to return any number of tokens from the multiplied investment to the Investor, but importantly, does not have to do so. The tokens are redeemed for actual money at the end of the experiment (see Methods).

Due to the information asymmetry between the two players in the HMTG, different moral strategies predict different decisions for the Trustee when the multiplier is ×2 or ×6. A guilt-averse Trustee, eager to match the Investor's expectations[7], should always return the number of tokens that were expected based on the Investor's belief in a fixed ×4 multiplier, irrespective of the actual multiplier employed on that trial. An inequity-averse Trustee, however, keen to ensure an even split[4,5], will instead base his decision on the total number of tokens he receives—which depends on the actual multiplier used—and ensure an equal division between the Investor and himself. A third expected moral strategy is greed, which simply predicts that the Trustee keeps as many tokens as possible. Importantly, our game also allows for identification of a fourth, context-based moral strategy, which we term moral opportunism. Here, we predict a Trustee would be inequity-averse in the ×2 condition but guilt-averse in ×6, thus always following a non-greedy moral rule, but one that is the most financially beneficial at any given time. At first glance, such an opportunistic strategy would appear peculiar, since it consistently minimizes neither guilt nor inequity, and indeed leaves the Trustee with fewer game tokens than a simple greed strategy. However, the moral opportunism prediction follows from the notion that some decision-makers might not follow context-independent moral heuristics, but rather decide flexibly which course of action in a given situation is both morally justifiable and maximally financially lucrative.

To more clearly distinguish between these various moral strategies, and to identify potential intermediate strategies, we developed a computational model to formalize the reciprocity motives of the second player (the Trustee) in the game. Integrating previous models of inequity aversion, guilt aversion, and greed, our Moral Strategy Model posits that the Trustee's utility results from a trade-off between financial self-interest (monetary payoff) and social preferences (Guilt/Inequity), weighted by a greed parameter (Theta; $\Theta$). We define the Trustee's payoff $\pi_2 = (I \times M_2 - S_2)/(I \times M_2)$, where $I$ is the Investor's investment amount, $M_2$ is the multiplier known only to the Trustee, and $S_2$ describes the Trustee's strategy (i.e., the amount of money to return in the game). We used previous formulations of inequity aversion, $\text{Inequity}_2 = ((I \times M_2 - S_2)/(10 - I + I \times M_2) - \frac{1}{2})$[2,4], and a nonlinear version of guilt aversion, $\text{Guilt}_2 = ((E_2 (E_1(S_2)) - S_2)/(E_1 (M_1) \times I))^2$[7,17], where $E_2(E_1(S_2))$ refers to the Trustee's second-order belief about the Investor's expectations of the Trustee's strategy and $E_1(M_1)$ refers to the Investor's belief about the multiplier (always ×4). To maximize generalizability of our model, we fixed these second-order expectations across participants by setting them to half the amount the Investor believes the Trustee has $(E_2(E_1(S_2)) = \frac{1}{2} \times E_1(M_1) \times I)$. Self-report data confirmed that this is an accurate reflection of the Trustees' average second-order expectations (see Supplementary Figure 1). On each trial, the social preference term in the Moral Strategy Model consists of either guilt aversion or inequity aversion, and by default (at Phi ($\Phi$) = 0) the model selects whichever of the two motives yields the smallest loss in utility. As a consequence of this structure, the model can accommodate the contextual preferences found in moral opportunism, as it allows the Trustee to ignore guilt in the ×2 condition and ignore inequity in ×6. If $\Phi$ deviates from 0, however, decisions are biased toward moral consistency in the guilt-averse ($\Phi < 0$) or inequity-averse ($\Phi > 0$) direction. The Trustee thus

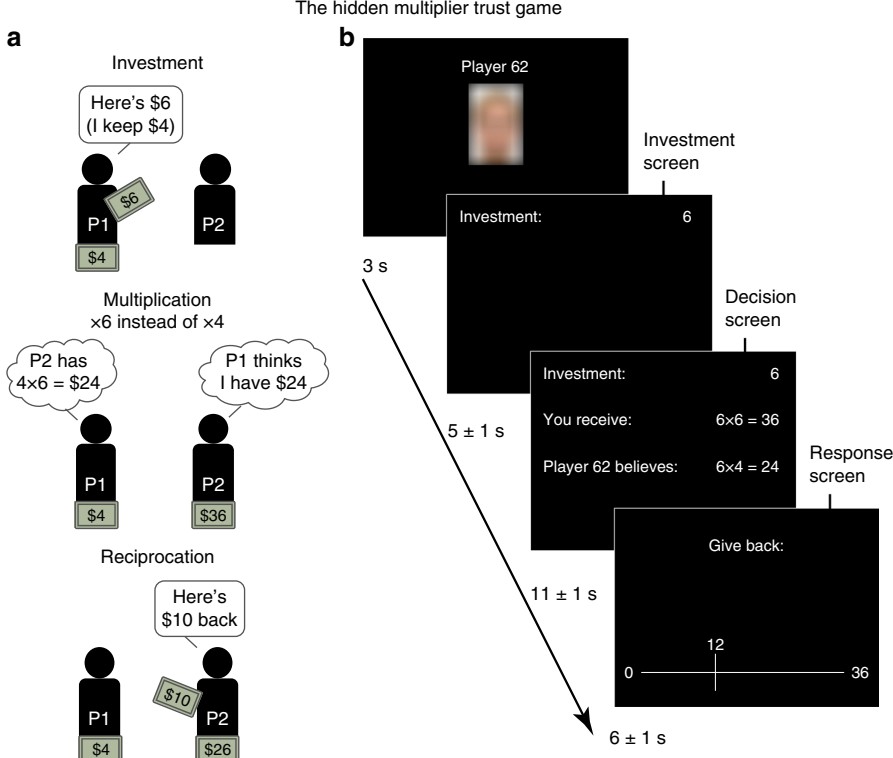

**Fig. 1** Task. **a** Schematic representation of the Hidden Multiplier Trust Game. P1 is the Investor; P2 is the Trustee. The participants in the current study always played as Trustee. **b** Trial timeline. The participant was instructed to make his decision during the Decision screen, and to use the Response screen only for carrying out the behavioral response

makes decisions that maximize the following utility function:

$$U_2 = \Theta \cdot \pi_2 - (1 - \Theta) \cdot \min\bigl(\text{guilt}_2 + \Phi, \text{inequity}_2 - \Phi\bigr) \quad (1)$$

It is important to note that the Moral Strategy Model cannot be estimated from behavior in a traditional Trust Game, where the behavioral patterns of guilt aversion, inequity aversion, and moral opportunism are aligned (see the ×4 condition simulations in Fig. 2b). Determining a participant's moral strategy therefore requires fitting one model to the participant's behavior across all conditions of the hidden multiplier task.

Overall, we find evidence that participants vary in their moral strategies when playing the HMTG. Fitting our Moral Strategy Model to participant behavior reveals that all four predicted moral decision strategies are present in our experimental sample. By linking the representational geometry of the computational model to the functional MRI data, we find evidence demonstrating that different moral strategies are associated with distinct neural activation patterns, even when they yield the same decision outcome. Moreover, we find evidence for context-dependent strategies. Morally opportunistic participants adaptively switch between guilt-averse and inequity-averse behavior, which corresponds to shifts in their brain activation patterns.

## Results

**Individual variation in decision strategies**. The Hidden Multiplier Trust Game successfully elicited reciprocity behavior in our sample, with 56 out of 57 Trustees choosing to return nonzero amounts to the Investors. However, participants used different strategies to decide how much money to return (Fig. 2a; see Supplementary Figure 2 for all participants' task behavior). All four hypothesized strategies were represented in our sample (Fig. 2a). To formally characterize these apparent strategy differences, we fit the Moral Strategy (MS) Model to each

participant's full set of behavioral responses in the Hidden Multiplier Trust Game. The Moral Strategy Model accurately described the different hypothesized moral strategies (Fig. 2b; Supplementary Figure 3), and captured task behavior significantly better than the unitary models of greed, guilt aversion, and inequity aversion, as determined by the Akaike Information Criterion (AIC): ΔAIC with respect to greed model = −229.33, $p < 0.001$; ΔAIC w.r.t. guilt aversion = −82.33, $p < 0.001$; ΔAIC w.r.t. inequity aversion = −11.24, $p = 0.021$ (Fig. 2c). These results match those obtained in a direct behavioral replication of this experiment ($n = 102$; see Methods), where model AIC was again lowest for the moral strategy model: ΔAIC w.r.t. greed = −220.36, $p < 0.001$; ΔAIC w.r.t. guilt aversion = −76.04, $p < 0.001$; ΔAIC w.r.t. inequity aversion = −13.39, $p = 0.010$ (Supplementary Figure 4A). The participants' model parameters were distributed throughout the model's two-dimensional parameter space (Fig. 2d), confirming the heterogeneity of moral strategies in our sample and the presence of intermediate strategies. This heterogeneity highlights the importance of studying inter-participant variation in moral decision-making, as averaging neural measurements over these 57 participants would likely obscure any brain processes specific to a single strategy.

Parameter recovery tests indicated that the parameters of the model were identifiable (correlation between true and recovered theta: $r = 1.00$, $p < 0.001$; phi: $r = 0.93$, $p < 0.001$; see Methods). To ensure that our model was not overfitting the data, we performed fivefold cross-validation within each participant's dataset to determine the model performance using unbiased parameter estimates (see Methods). Overall, we observed a high degree of model accuracy in held-out data (mean squared error per trial = 5.37, mean $r^2 = 0.86$; one-sample $t$ test on $r$ values: $t(55) = 66.1$, $p < 0.001$). These results confirm that our model was able to strongly predict participant's trial-level behavior, and

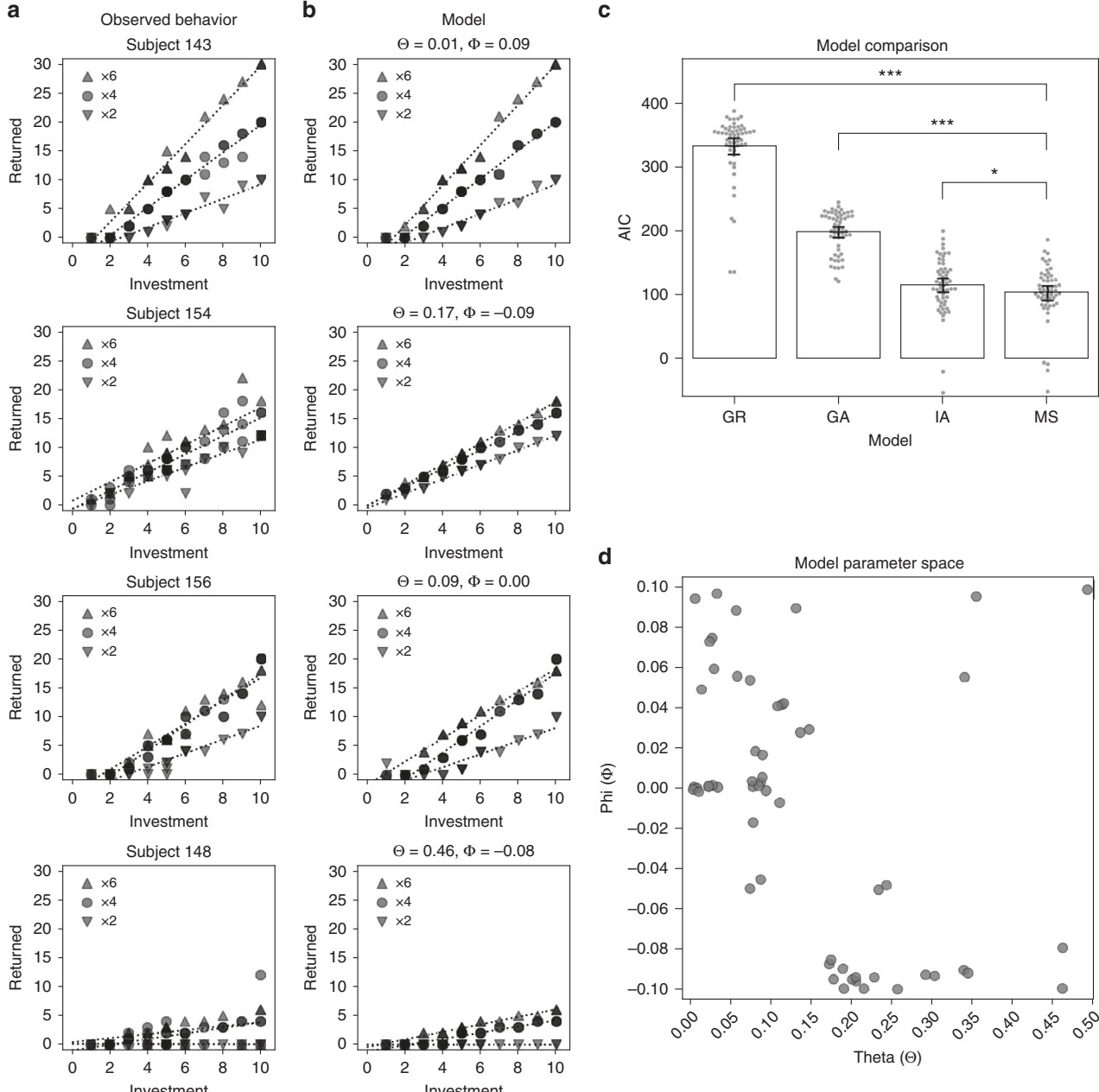

**Fig. 2** Between-subject strategy variety in the Hidden Multiplier Trust Game. **a** Observed behavior of four example participants, displaying different moral strategies (top to bottom: inequity aversion—143, guilt aversion—154, moral opportunism—156, greed—148). Data points are semi-transparent: darker points represent multiple observations. Dotted lines are regression lines per multiplier condition. **b** At different parametrizations, the computational model captures the strategy variety observed in **a**, **c** Model comparisons. Moral Strategy (MS) Model best fits the observed behavior. AIC Akaike Information Criterion, GR greed, GA guilt aversion, IA inequity aversion, MS Moral Strategy Model. ***$p < 0.001$; *$p < 0.05$; $p$-values from non-parametric paired-sample permutation tests. Error bars represent bootstrapped 95% confidence intervals. **d** Two-dimensional continuous parameter space of the Moral Strategy Model, with the participants scattered by best-fitting model parameters

indicate that participants were internally consistent in their moral strategy over time (Supplementary Figure 6), which allows us to infer a participant's moral strategy even in the ×4 condition of the Hidden Multiplier Trust Game, where the behavior of inequity aversion, guilt aversion, and moral opportunism is the same.

**Individual variation in brain activity reflects differences in strategy.** Having established that participants exhibit a variety of reciprocity decision strategies, we next examined how these different strategies might be implemented in the brain.

Standard analytic approaches are not well suited for answering this question. Ideally, we could map the model predictions directly onto brain responses at the trial level using a model-based fMRI approach.[18] However, our model requires all of a participant's decision behavior (i.e., across the three contexts) to identify their specific moral strategy, and therefore moral strategy measurements only exist at the participant level. Furthermore, standard contrast-based analyses of participant-level parameter estimates derived from general linear model (GLM) analyses are unable to provide interpretable inferences about the brain when participants differ on two separate

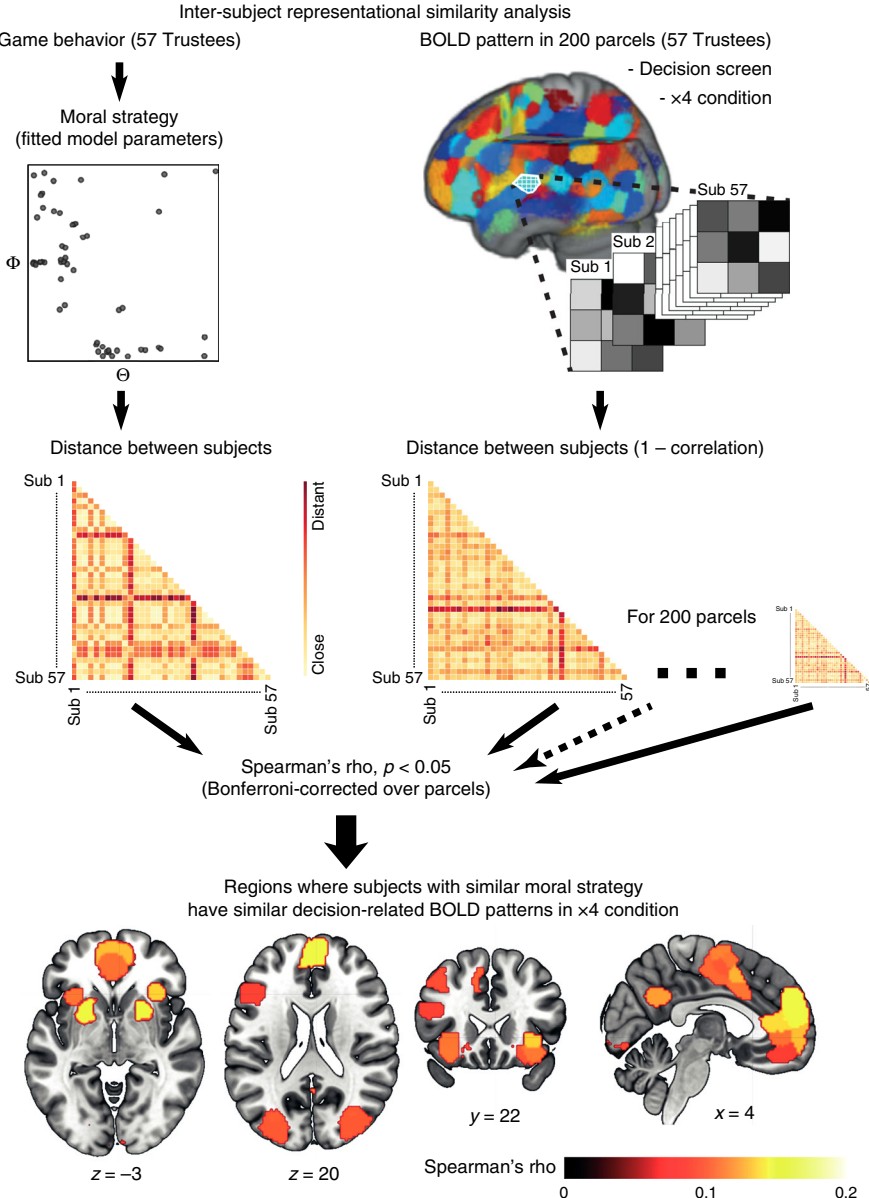

**Fig. 3** Inter-subject representational similarity analysis. Inter-subject RSA revealed brain parcels where players with similar moral strategy had similar BOLD patterns in the Decision screen of the ×4 condition. Since this condition yielded identical behavioral predictions for guilt-averse, inequity-averse, and morally opportunistic participants, neural pattern differences likely reflect the psychological computations underlying these moral strategies. Brain slice numbers represent coordinate in MNI space

continuous variables (i.e., the model parameters). Therefore, we instead employed multivariate pattern analysis, which can map multidimensional psychological states or processes to specific brain regions[19–24]. Specifically, the second moment of multi-voxel brain representations across stimuli can reflect meaningful differences in how stimuli are psychologically organized, as demonstrated in representational similarity analysis (RSA)[25–27] For example, the psychological organization of moral judgments, such as harm versus impurity judgments, can be estimated from multivariate activity patterns in the mentalizing network[28].

Here, we extend these analytic advances to a between-subjects design using inter-subject representational similarity analysis (IS-RSA; Fig. 3). This method combines two developments in neuroimaging analysis: the geometric mapping of relationships between stimulus features, as proposed in RSA[26] and the similarity of computations in a specific brain region across participants, as proposed in inter-subject connectivity[29]. IS-RSA

allowed us to map variations in brain processes associated with HMTG decisions directly onto our Moral Strategy Model, effectively testing whether multi-voxel activity patterns associated with reciprocity decision-making are similar for participants who decide in a similar way (and dissimilar for participants with a dissimilar decision strategy). To do this, we first created a geometric representation of the Moral Strategy Model's parameter space by computing the Euclidean distance between all pairs of participants. We then searched for brain regions that showed a similar representational geometry to this distance measure in terms of the multi-voxel activity pattern correlations[25] between each pair of participants during the decision screen of the task. To reduce the search space in the brain while performing this computation, we used an a priori 200-parcel whole-brain parcellation based on meta-analytic functional coactivation of the Neurosynth database[30], and we identified parcels that survived Bonferroni correction (i.e., $p < 0.00025$). We carried out this

analysis specifically in the ×4 condition of the HMTG, as here the predicted behavior in this condition for guilt aversion, inequity aversion, and moral opportunism is identical, which means that between-subject comparisons are maximally controlled for decision output and reward.

We observed significant inter-subject representational similarity effects in 27 brain parcels, including the ventral and dorsal medial prefrontal cortex (MPFC), dorsal anterior cingulate cortex (dACC), bilateral anterior insula (bilateral AI), bilateral putamen, bilateral premotor cortex, bilateral angular gyrus, and left DLPFC (Fig. 3). These results, which were specific to the decision phase of the HMTG task (Supplementary Figure 7), indicate that decision-related activity patterns in these regions were more similar between participants that share a similar moral strategy for reciprocity decisions than between participants who differed in their strategy. The degree of similarity is directly proportional to the distance between the participants in the model parameter space. Since predicted choice output in the ×4 condition is identical between IA, GA, and MO strategies, these regions are likely involved in the psychological computations that underlie the various strategies. These results provide converging evidence with previous studies using mass univariate fMRI analysis methods (e.g., AI, MPFC, and DLPFC)[13,14,31,32], but importantly our paradigm additionally allows us to evaluate the degree to which these regions might be *selectively* processing computations relevant to a specific moral strategy.

**Classifying participants' strategies based on model parameters.**
To test whether any brain region was selectively associated with inequity aversion, guilt aversion, or moral opportunism, we first identified the groups of participants who most strongly expressed these strategies in their reciprocity decisions. To this end, we divided the parameter space of the model into four moral strategy zones using a purely model-driven clustering approach. We simulated Trustee behavior datasets from the model by varying the two free parameters (theta and phi) within the parameter bounds, and then used hierarchical clustering to group the simulations based on similarity, as parameters that yield similar behavioral predictions should be associated with the same strategy (Fig. 4a, colored zones; see Methods). The strategy clustering was defined solely from model simulation, without relying on any experimental data, ensuring that the cluster boundaries are not biased by the distribution of strategies we observed in our sample of participants.

Each participant's dominant moral strategy was then determined by the position of their model parameters within the theoretically defined strategy boundaries on the theta–phi plane. This grouping method yielded 24 inequity-averse (IA), 5 guilt-averse (GA), 21 morally opportunistic (MO), and 7 greedy (GR) participants. This distribution of moral strategies in our sample corresponds strongly to the distribution obtained in a direct behavioral replication of this experiment (cosine similarity $r = 0.96$, $n = 102$; see Supplementary Figure 4B, C). Mean Trustee behavior of the four groups is visualized in Fig. 4b and approximates the theoretical predictions of the four moral strategies (see Supplementary Figure 5 for individual subject behavior grouped by strategy). Importantly, the four strategy groups did not differ on age, gender, or tendency to experience guilt (Supplementary Table 1). Consistency of choices (as indexed by model error) also did not differ between the groups, and was approximately equal across the parameter space of the model (Supplementary Table 1; Supplementary Figure 8). Inequity-averse participants scored higher than greedy and morally opportunistic participants on Social Value Orientation (SVO; see Methods), where higher scores correspond to inequity-averse

preferences and lower scores to greed[33]. Thus, the behavior of inequity-averse and greedy participants on the HMTG generalized to the SVO task, which lends construct validity to our task, computational model, and method of grouping participants. Finally, the three groups with identical predictions in the ×4 condition (IA, GA, and MO) did not differ on actual choice behavior in this condition (linear mixed-effects regression, main effect of group on number of tokens returned: $F(2,47) = 2.61$, $p = 0.084$; Supplementary Figure 9) and accordingly did not differ on number of tokens earned in this condition (one-way analysis of variance (ANOVA) on sum earned in ×4, effect of group: $F(2,47) = 2.51$, $p = 0.092$).

**Decision strategies are associated with distinct brain patterns.**
If a brain region is selectively involved in processing a specific moral strategy, we would expect this region to exhibit a specific multi-voxel activity pattern exclusively in the participants employing this moral strategy, i.e., one that is distinct from participants relying on other decision strategies. This intuition is operationalized by a measure commonly used to evaluate unsupervised machine-learning models, which we call the "cluster strength score"[34,35] (see Methods). For each participant, this metric indexes the pattern similarity to other participants who employ the same moral strategy, relative to the pattern similarity to all other participants. We used a sign permutation test over a given strategy group's cluster strength scores to test whether this strategy was significantly associated with a brain parcel identified in IS-RSA. This test implicitly controls for the size of the group. We limited this analysis to the ×4 condition of the task, and excluded the greedy subjects, to rule out behavioral differences between the strategy groups.

We found that the guilt-averse subjects shared a unique activity pattern in the bilateral anterior insula, bilateral putamen, DMPFC, and left DLPFC (Fig. 4c). In contrast, inequity-averse subjects shared an exclusive pattern in bilateral AI, VMPFC, dACC, supplementary motor area, and bilateral superior occipital cortex. Moral opportunists shared a common pattern in bilateral superior parietal cortex and dACC. These findings indicate that information pertaining to specific moral strategies is encoded in unique patterns of multi-voxel activity in specific brain regions, with patterns and regions being consistent across participants using the same strategy. This is particularly interesting in light of the fact that all three of these moral strategies involve the returning of the same amount of money in this ×4 context, suggesting that these regions are implementing computations unique to the underlying moral decision strategies.

**Consistent brain patterns across conditions.** If the activity patterns observed to be exclusive to guilt aversion, inequity aversion, and moral opportunism in the ×4 condition indeed reflect neural computations that are meaningfully related to the associated decision strategies, we would also expect these activity patterns to be stable across task conditions. That is, each participant's activity pattern in the ×4 condition should be similar to the patterns of other participants using that strategy in the ×2 and ×6 conditions, and conversely dissimilar to the ×2/×6 patterns of subjects using any other strategy. To test this, we calculated the degree to which the spatial pattern in the ×4 condition for a given participant was more similar to other participants using the same strategy in the ×2 and ×6 condition, as compared with participants employing other strategies (see Methods). The results of this analysis (Fig. 4d for generalization to ×2; Fig. 4e for generalization to ×6) were largely consistent with the ×4 condition pattern clustering results.

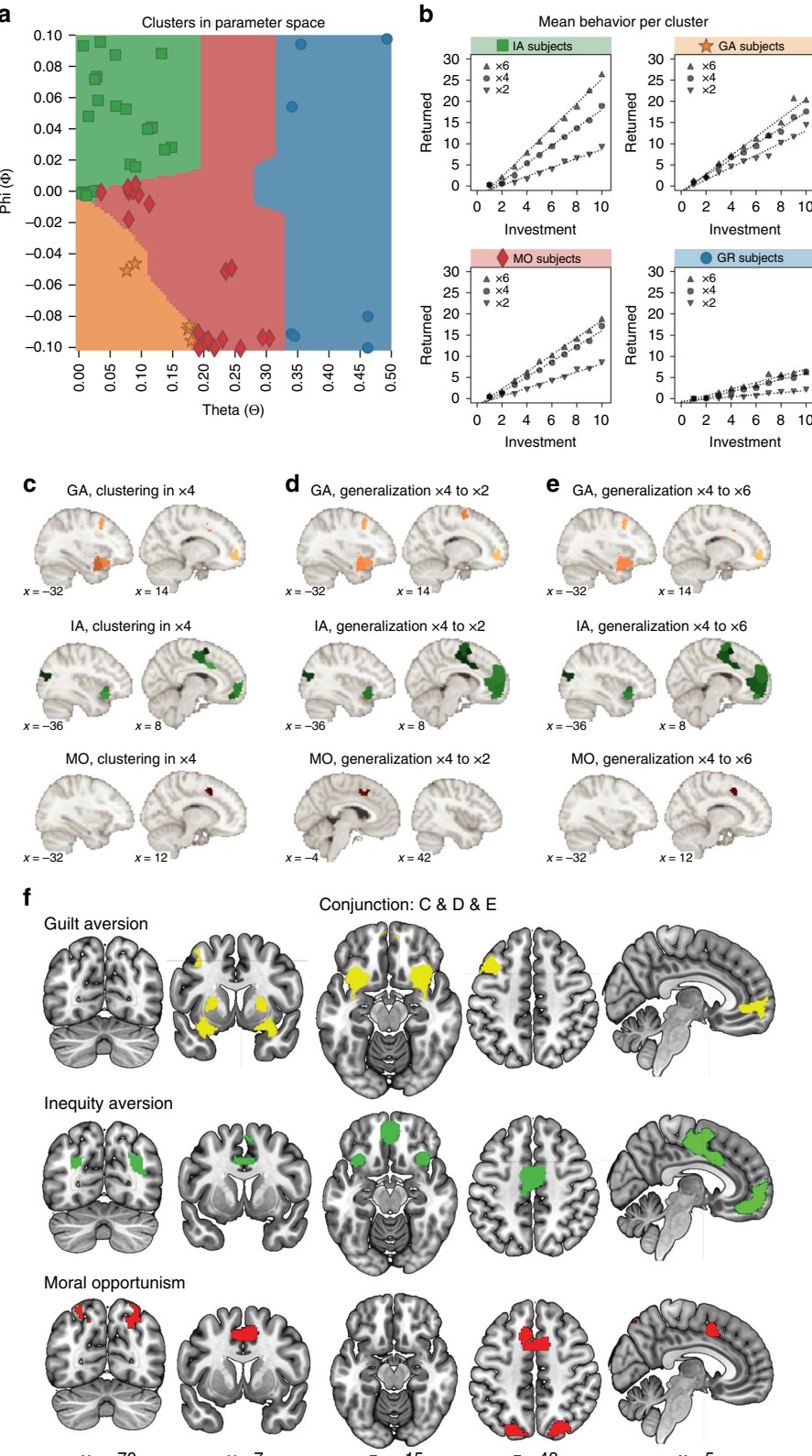

**Fig. 4** Strategy-linked activity patterns replicate over participants and contexts. **a** Model-driven clustering of the parameter space by moral strategy based on similarity between model simulations. **b** Mean reciprocity behavior in each of the four strategy groups reflects the theoretical predictions of IA, GA, MO, and GR. IA inequity-averse, GA guilt-averse, MO morally opportunistic, GR greedy. **c** Multi-voxel activity patterns that replicate over subjects within moral strategy group in the ×4 condition (i.e., patterns generalize over subjects within group). **d** Parcels where strategy-specific ×4 activity patterns generalize across subjects to ×2. **e** Parcels where strategy-specific ×4 activity patterns generalize across subjects to ×6. **f** Conjunction of **c**, **d**, and **e**. Activity patterns in these parcels clustered within the respective strategy group in the x4 condition, and generalized to ×2 and ×6

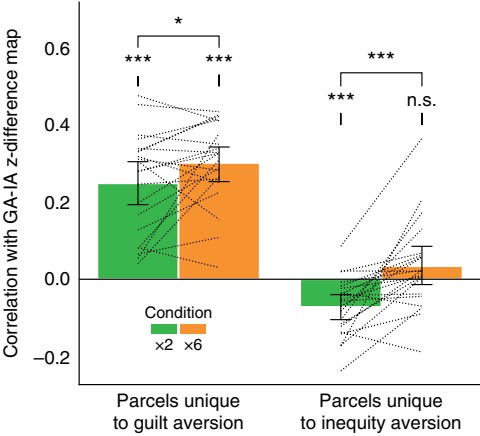

**Fig. 5** Moral opportunists shift between guilt-averse and inequity-averse brain representations. Moral opportunists most strongly expressed the guilt aversion pattern in the condition where they employed guilt aversion computations (×6), and vice versa for inequity aversion in ×2. Bar height represents the mean over all moral opportunists; error bars represent bootstrapped 95% confidence interval; dashed lines connect pairs of data points from the same participant. *$p < 0.05$; ***$p < 0.001$; n.s. not significant; $p$-values from one-sample and paired-sample $t$ tests

Since we were specifically interested in finding brain parcels where activity patterns generalized across both participants and conditions within a given moral strategy group, we identified regions where the conjunction was significant across analyses (Fig. 4f). These converging "strategy maps" indicate that the guilt aversion strategy involved the ventral surface of the bilateral AI, bilateral putamen, MPFC, and left DLPFC; inequity aversion involved bilateral AI, VMPFC, dACC, and bilateral intraparietal sulcus; and moral opportunism involved bilateral superior parietal cortex (SPC) and dACC. These regions replicate earlier findings on the neural correlates of social preferences.[9–13] The insula, putamen, DLPFC, and VMPFC have previously been associated with guilt aversion[13,14], and the SMA and VMPFC have been found in previous studies examining inequity aversion ([10,11,36,37], among many others). While both these moral strategies engage the AI via distinct multivariate patterns, guilt aversion does so more extensively on the ventral surface of this region. Interestingly, the only regions where moral opportunists showed consistent and exclusive activity patterns are regions associated with cognitive control (SPC and dACC[38–40]), which may be related to the cognitive resources required to switch between competing strategies.

**Moral opportunists shift brain patterns**. If decision-related activity patterns in the "strategy maps" (Fig. 4f) indeed reflect the unique psychological computations involved in inequity aversion (IA) and guilt aversion (GA), then we should be able to predict participant strategies out of sample based solely on their brain activity. The morally opportunistic (MO) groups are well suited for this type of confirmatory analysis, as their behavioral strategy shifts from IA to GA depending on the trial context. We hypothesized that the MO players should express the GA and IA patterns most strongly in the conditions where they selectively use the associated computations in their decision-making, that is, in the ×2 condition for inequity aversion and the ×6 condition for guilt aversion.

To test this prediction, we computed the similarity of the moral opportunists' activity patterns to the GA–IA pattern difference map for each parcel and condition (see Methods). A positive similarity score would indicate that an MO participant was more

similar to GA than IA, and vice versa. Results showed that, across all parcels, the mean similarity of MO participants to the GA–IA difference map was indeed significantly greater in the ×6 condition than in ×2 (mean correlation difference $\Delta r = 0.077$; paired-samples $t$ test: $t(20) = 4.37$, $p < 0.001$). This confirms our hypothesis that the morally opportunistic participants expressed the guilt-averse and inequity-averse activity patterns most strongly when they used the associated computations in their decision-making.

To rule out the possibility that this effect was driven by just one of the two patterns' being upregulated in the corresponding condition, we additionally tested the GA–IA pattern similarity in brain parcels uniquely associated with either guilt aversion or inequity aversion (i.e., the non-overlapping parcels of the GA and IA strategy maps from Fig. 4f). Interestingly, we found that MO activity patterns in IA-specific parcels were more similar to IA than to GA in ×2 (mean $r = -0.071$, one-sample $t$ test $t(20) = -4.30$, $p < 0.001$), but not significantly so in ×6 (mean $r = 0.034$, $t(20) = 1.32$, $p = 0.20$) (Fig. 5). In these parcels, pattern similarity to the GA–IA difference map was significantly higher in ×6 than in ×2 ($\Delta r = 0.11$, paired-samples $t(20) = 4.31$, $p < 0.001$). Conversely, MO activity patterns in GA-specific parcels were more similar to GA than to IA (×2: mean $r = 0.25$, one-sample $t$ test $t(20) = 8.57$, $p < 0.001$; ×6: mean $r = 0.30$, $t(20) = 12.9$, $p < 0.001$), and more so in ×6 than in ×2 ($\Delta r = 0.05$, paired-samples $t$ test $t(20) = 2.15$, $p = 0.044$). These results confirm that moral opportunists can express both the GA and IA activity patterns, alternating between the two, depending on the condition of the task.

The moral opportunists thus not only used different behavioral strategies according to the multiplier used; they also differentially expressed the activity pattern found in the associated moral strategy group (GA or IA) depending on the task condition. Moreover, the specific strategy used by a Moral Opportunist in the ×2 and ×6 contexts could be classified with 90.5% accuracy using only the relative similarity score to the GA/IA brain patterns, indicating that 19 out of 21 of Moral Opportunists had a neural activation pattern that was most similar to GA in the ×6 and to IA in the ×2 context. This provides strong evidence using an independent sample that the patterns we observed in the inequity-averse and guilt-averse participants directly reflect the moral strategy computations carried out in the corresponding strategy maps, confirming that these patterns capture meaningful signal related to the underlying psychological process.

## Discussion

In this paper, we have presented experimental evidence illustrating several distinctive decision strategies when reciprocating another person's trust. These variations in moral strategies were computationally characterized using a utility model that integrates previous formulations of guilt and inequity aversion. By leveraging the between-participant differences captured in the two-dimensional parameter space of this model, we mapped psychological computations corresponding to guilt and inequity aversion to specific parts of the human brain.

Importantly, our inferences about moral strategies required the use of a task in which different social preference models yield different behavioral predictions (i.e., The Hidden Multiplier Trust Game). This variant of the canonical Trust Game allowed us to disentangle neural processes related to different motivational signals, which would have been conflated in the traditional version of the task. Computationally characterizing moral strategies at the individual level allowed us to draw inferences about motivational differences even in the task condition where the behavioral predictions for guilt aversion, inequity aversion, and

moral opportunism were the same (the ×4 condition). However, these participant-level inferences required a different analytical approach from traditional model-based fMRI. A standard contrast-based GLM analysis is not well suited for examining our effects of interest, but IS-RSA allowed us to map the complex geometry of participants in multidimensional model space onto individual differences in neural signals. By analyzing how participants cluster in high-dimensional activity pattern space, we demonstrated that the psychological computations underlying guilt aversion and inequity aversion are implemented in different sets of brain regions. The guilt-averse moral strategy was associated with the AI, putamen, MPFC, and left DLPFC. These findings support the previously proposed idea that computations in the AI facilitate a guilt response when not living up to the expectations of another person[13,36]. Social expectations themselves may be computed in Theory of Mind regions, such as MPFC[41,42], where we also found evidence for guilt aversion-specific computations. In contrast, inequity aversion computations were mapped onto the AI, VMPFC, and dACC. The AI[10] and VMPFC[11] have been linked to this social preference before, while the involvement of the dACC may relate to this region's role in monitoring task performance[43] and tracking one's position in a social hierarchy[44]. If the inequity aversion motive revolves around minimizing payoff differences between self and other, the medial frontal cortex (including VMPFC and dACC) is well suited to carry out this computation, due to its role in self-referential payoff processing[11,45,46]. Interestingly, the guilt aversion and inequity aversion strategies overlap in AI, but using distinct multi-voxel patterns of activation.

We also report a new strategy observed in participants, moral opportunism. This group did not consistently apply one moral rule to their decisions, but rather appeared to make a motivational trade-off depending on the particular trial structure. This opportunistic decision strategy entailed switching between the behavioral patterns of guilt aversion and inequity aversion, and allowed participants to maximize their financial payoff while still always following a moral rule. Although it could have been the case that these opportunists merely resembled GA and IA in terms of decision outcome, and not in the underlying psychological process, a confirmatory analysis showed that the moral opportunists did in fact switch between the neural representations of guilt and inequity aversion, and thus flexibly employed the respective psychological processes underlying these two, quite different, social preferences. This further supports our interpretation that the activity patterns directly reflect guilt aversion and inequity aversion computations, and not a theoretically peripheral "third factor" shared between GA or IA participants. Additionally, we found activity patterns specifically linked to moral opportunism in the superior parietal cortex and dACC, which are strongly associated with cognitive control and working memory[38–40]. We speculate that processes relevant for switching strategies may have resulted in moral opportunists consistently recruiting these regions.

This study demonstrates how RSA can be used to measure between-subject differences in neural function. An advantage of this method compared with traditional model-based fMRI is that IS-RSA does not require a strong prior about the algorithmic implementation of the psychological computation of interest[47,48]. Because the inference in IS-RSA is based on inter-subject similarity, it does not require specifying how the computations are directly encoded by a specific voxel. Encoding models of the complex computations of social cognition, emotion, and the prefrontal cortex more broadly have substantially lagged behind models characterizing how stimulus features are encoded by the sensory cortex. IS-RSA could therefore be useful to researchers aiming to map psychological computations onto the brain while remaining agnostic about the neural algorithm. Here, we demonstrate that the brain patterns can be used as an intermediate representation of this process, but are not themselves directly interpretable.

More generally, our approach allowed us to leverage endogenous between-participant differences in psychological processing of the task at hand, while traditional analysis methods would have required us to average measurements across participants with potentially vast differences in task interpretation. As such, our methods open up the possibility of treating participant-level variation as signal instead of noise, to avoid averaging out key functional features of the human brain[49]. In this way, IS-RSA can also facilitate the use of more ecologically valid experimental paradigms, by allowing participants to vary in how they interpret the task.

One major strength of our approach is that participants freely decided which strategy to employ while making their reciprocity decisions, in contrast to prior research where participants were instructed to reason in a particular way (e.g., ref. [15]). An obvious downside of this approach is that we could not control the relative frequencies of moral strategies in our dataset, which contained a relatively low number of purely guilt-averse participants. However, our analytic approach accounted for this in several ways. First, we defined the participant clustering based purely on the computational model, which meant the relative prevalence of various strategies in our sample did not bias the clustering solution. Second, the inter-subject RSA and brain-space clustering analyses were based on measurements of pairwise similarity between pairs of participants, which is effectively using an ordinal rather than cardinal scaling on a multidimensional space akin to preference learning. Third, we used within-group permutation tests to assess the statistical significance of brain activity pattern clustering, which implicitly controls for differences in group size. Fourth, we developed several converging lines of evidence—for a brain parcel to be included in the guilt aversion "strategy map", it had to show significant clustering of GA participants in the ×4 condition, as well as significant generalizability of those patterns to the ×2 and ×6 conditions, which reduces the likelihood of false positives. Finally, we used an out-of-sample test to demonstrate that the activity patterns employed by guilt-averse and inequity-averse participants were functionally linked to behavior using the moral opportunists. Thus, we believe our results are robust to the unequal sample sizes of each strategy type. The low prevalence of pure guilt aversion in our sample may be helpful for improving psychological game theory[50] as it suggests that some guilt-averse behavior in past research may in fact have been motivated by inequity aversion, and that guilt-averse preferences may be context-dependent (e.g., for moral opportunists in our task).

A strength of our analysis is that we grouped participants based on an a priori clustering of the parameter space of the computational model. This means that the cluster boundaries in parameter space are invariant to any given sampling of participants from the broader population. Interestingly, our data indicate a relatively high density of participants around cluster boundaries, where some participants are closer to a cluster boundary than to other participants in the same cluster. We do not believe that this should substantially impact our interpretation of the results. First, our model simulations indicate (Supplementary Figure 3) that unit increases in parameter space do not linearly correspond to changes in predicted behavior. For example, if the theta parameter is high, then the behavior produced by large variations in phi are negligible, while small changes in phi produce large changes in behavior when theta is low. In addition, there are multiple combinations of the parameters that can yield very similar predicted behaviors, and often these parameter combinations are far in parameter space but within a cluster boundary.

Second, we are primarily interested in connecting the representational geometry of this model space to that of brain space. This means that we are more interested in the relative ordinal distance between individuals in the two feature spaces than their absolute cardinal distance in the parameter space units. This is evaluated in the primary IS-RSA, which is agnostic to the clustering boundaries and uses rank-ordered (Spearman) correlations to link the two feature spaces. The cluster boundaries merely provide a heuristic to aid in interpreting the brain findings. We hope to better understand these nuances of the model as well as cultural specificity of the moral strategies in future work.

The stability of individual participants' moral strategies over trials raises several interesting questions for future work. First, how stable are moral strategies across tasks? If participants use the same moral strategy throughout different contexts, tasks, and time points, this might reflect a trait-like "moral phenotype" anchored in participants' enduring social preferences. This motivational trait might underlie the behavioral phenotypes previously observed across economic games[51,52]. However, given that there are reports of low correspondence between laboratory and field behavior in the same participants[53] (but see ref. [54]), more work is needed to evaluate this possibility. Second, do decision strategies inferred from behavior correspond to participants' subjective experiences? Although there is evidence that participants are generally unable to accurately report their internal cognitive processes[55], it would be interesting to explore the subjective reasoning of participants whose strategy lacks internal consistency (e.g., moral opportunists). Third, what leads to the development of a mix of moral strategies in a population? In many human social interactions, moral opportunism may be the most adaptive strategy, as it allows participants to maximize payoff while still being able to justify their decision to others—a key goal of moral reasoning[2,56,57]. However, there may be psychological or cultural boundary conditions to this strategy that drive some individuals toward moral consistency.

Taken together with other recent work on between-subject neural clustering[58], our observations suggest that patterns of brain activity align when people have a similar experience or interpretation of an event. For example, a visual percept shared across people is likely to reflect a similar neural representation in early sensory cortex. Reflecting on the moral debates that divide our societies today, this remarkable property of the human brain suggests that our political adversaries are not obtuse, naïve, or ignorant, as we may be inclined to believe, but rather may be reasoning about moral dilemmas in a fundamentally different way.

## Methods

**Participants**. Sixty-six participants were recruited from the Nijmegen student population through a web-based registration tool. Students of psychology or economics were excluded from participation, as they were potentially familiar with game theory or the Trust Game. All participants were screened for significant health or neurological problems and had normal or corrected-to-normal vision, and all gave written informed consent before the start of the experiment. Nine participants were excluded from the analysis because of excessive head movements in the MRI scanner, misunderstanding of the task, disbelief in the task, or technical issues. Fifty-seven participants (mean age = 21.3 ± 2.1 years, 39 women and 18 men) remained. The experiment was approved by the local ethics committee (CMO Arnhem–Nijmegen, the Netherlands).

**Experimental procedures**. The experiment consisted of a single session. The participant was first seated in a behavioral lab space to complete screening and informed consent forms and to read the task instructions. To avoid biasing game behavior, the Trust Game was always referred to as "Investment game", the Investor as "player A", and the Trustee as "player B". The participant was instructed that he/she would play 80 single-shot trials in the role of player B with 80 anonymous players A, each of whom had previously participated and consented to have their data used here. Participants were instructed that they would be paid based on their response to one randomly selected trial at the conclusion of the

experiment, and that this trial would be financially consequential for the Investor too. The choice behavior of the Investors was drawn from an actual Trust Game dataset previously collected with the same task parameters (multiplier always ×4[13]), but the randomly selected trial was only financially consequential for the participant. To enhance the plausibility of the task, participants were asked to make their own investment decision as player A to an anonymous player B, and were told that they would be contacted and paid if their investment decision was used in a similar future experiment. After the instructions, participants' photos were taken for (blurred) use in the possible future Trust Game experiment. While undergoing fMRI of the brain, participants played 80 trials of the Hidden Multiplier Trust Game (HMTG).

**Task**. The Hidden Multiplier Trust Game (HMTG; Fig. 1) is a variant of the regular Trust Game (also known as Investment Game[16]) with one important difference: the multiplier varies between ×2 (25% of trials), ×4 (50%), and ×6 (25%). Only the Trustee knows the actual multiplier. The Investor believes the multiplier is always ×4, and the Trustee knows about the Investor's ignorance. The resulting information asymmetry allows us to probe the Trustee's motivations to reciprocate in the game: guilt aversion predicts that the amounts sent back by the Trustee do not differ between multiplier conditions, whereas inequity aversion predicts that Trustees are sensitive to the changing multiplier. Moral opportunism predicts that Trustees are guilt-averse in ×6 and inequity-averse in ×2. The investments and multipliers were assigned to the 80 trials such that the distribution of investments was highly similar between the multiplier conditions (×2, ×4, and ×6; Supplementary Figure 10) and identical across participants. In 4 out of 80 trials (5% of each multiplier condition), the Investment was 0, so the Trustee could not respond. These non-informative trials were excluded from computational modeling. The trials were presented in a different random order to each participant. On the first screen of each trial, the Trustee was presented with the participant number of the Investor and a blurred picture of a face to strengthen the Trustee's social experience in the task.

**Stimulus presentation**. The task was divided into two runs of 40 trials each. Each run lasted around 18 min with 30 additional TRs of fixation at the beginning, which were used to compute the combining weights for the four echoes in our multi-echo fMRI sequence. Before the first run, there was a left-handed finger tapping task and a calibration procedure for eye tracking. Between the runs, the participant was allowed to take a break for as long as he/she wanted. At the end of the scanner session, a T1-weighted anatomical scan was made (see "fMRI data acquisition"). All stimuli were presented using PsychToolBox 3.0.11 (www.psychtoolbox.org) in MATLAB 2013a (Mathworks, Natick, MA, USA) onto a screen at the back of the scanner bore, which the participant could view using a mirror mounted onto the head coil. The participant responded using the leftmost two buttons on a four-button curved response box (Current Designs, Philadelphia, PA, USA) in the right hand. These buttons moved the slider on the decision screen left and right in increments of 1 token or 10% of the slider range (whichever was greatest, to increase the speed of movement on the slider[13]). The slider ranged from 0 to [investment × multiplier]. The starting point of the slider was randomly selected on each trial, ensuring that the number of button presses was orthogonal to the number of tokens selected.

**fMRI data acquisition**. Functional magnetic resonance imaging was performed at the Donders Centre for Cognitive Neuroimaging in Nijmegen, The Netherlands, using a 3-Tesla head-dedicated MRI system (Skyra; Siemens Medical Systems). T2*-weighted functional MR images were acquired using a 32-channel head coil and a multi-echo pulse sequence (224 -mm field of view (FOV); 64 × 64 matrix; 90° flip angle; 2250 ms repetition time (TR); echo times (TE) 9.4 ms, 20.6 ms, 32 ms, and 43 ms). Thirty-five ascending slices were acquired (slice thickness 3.0 mm; slice gap 0.5 mm; voxel size 3.5 × 3.5 × 3.0 mm), covering the whole brain except the cerebellum. A high-resolution T1-weighted image was acquired using an MPRAGE sequence (192 sagittal slices; TR 2300 ms, voxel size 1 × 1 × 1 mm). To minimize head movement, soft adhesive tape was placed across the participant's forehead immediately before image acquisition started. In accordance with safety regulations, the participant wore earplugs during the experiment and had access to an alarm button.

**Additional measures**. After the scanner session, the participants were brought into the behavioral lab to complete several computerized tasks and paper questionnaires. First, they were asked to rate, for each HMTG trial in the scanner, how guilty they felt about the number of tokens they had returned to the Investor, and how guilty they would have felt if they had returned a randomly selected alternative number of tokens. Ratings between 1 and 7 were measured on a continuous, computer mouse-controlled slider. Next, the participants completed a questionnaire on their beliefs about the Investor's expectations at each possible investment in the HMTG. We used these self-reported second-order expectations to check that the model we used as a proxy for second-order expectations in our model (on each trial: 2 × investment) was accurate (see Supplementary Figure 1). The participant also completed computerized versions of the Social Value Orientation (SVO) task (slider version, incentivized; adapted from[59]), and the Guilt

Inventory, which measures an individual's propensity to guilty feelings in three categories (trait guilt, state guilt, and moral standards; see ref. [60]). Finally, the participant completed a demographics questionnaire on paper and answered several reflective questions about the experiment. One of these questions was "Do you think that the participants with whom you played in Task 1 will be happy if the round you played with them is selected for payment?" We used the participant's answer on this question to test (dis)belief in the financial consequentiality of the task for the Investors. Two participants were excluded from the analysis based on this question.

**Participant payment**. Several days after the experiment, one of the 80 HMTG rounds was randomly selected for the payment. The participant received the number of tokens earned (Investment × multiplier−Amount returned) in this round, converted to euro using an exchange rate of €0.40 per token. This amount was added to the earnings from the incentivized SVO task and a €29 base fee. An administrator not involved in the study electronically transferred the total amount to the participant several weeks after the experiment. The participants had been informed about the payment procedures at the start of the experiment.

**Behavioral data analysis**. Behavioral analyses were carried out in Python version 2.7.15 (Python Software Foundation), using the *Scipy* package version 1.1.0, unless noted otherwise below.

**Computational modeling**. The Moral Strategy (MS) Model (Eq. 1) was fit to each participant's behavioral data by varying the free parameters (theta and phi) within the parameter bounds ($0 \leq \Theta \leq 0.5$ and $-0.1 \leq \Phi \leq 0.1$), and minimizing the sum of squared error between the model's behavioral prediction and actual behavior over the 76 trials with nonzero investment, using the *least_squares* routine in *Scipy*. To avoid ending the fitting procedure in a local minimum, the model fitting algorithm was initialized at 10,000 random points in theta–phi parameter space for each participant. In case of a tie in model fit between two or more iterations of the fitting procedure, the first occurrence of the best model fit was selected as a winning model.

We compared the predictive accuracy of our model to that of its component models, including greed (Eq. 2), inequity aversion (Eq. 3)[4], and guilt aversion (Eq. 4)[7]. In each of the following formulations, $U_2$ refers to the Trustee's utility, and $\pi_2$ is the Trustee's payoff, defined as $\pi_2 = I \times M_2 - S_2$. $I$ is the Investor's investment amount, $M_2$ is the multiplier known only to the Trustee, $S_2$ describes the Trustee's strategy (i.e., the amount of money to return in the game), $E_2 E_1(S_2)$ refers to the Trustee's second-order belief about the Investor's expectations of the Trustee's strategy, and $E_1(M_1)$ refers to the Investor's belief about the multiplier (always x4). Theta ($\Theta$) is a greed parameter that weights social preference (inequity in Eq. 3 and guilt in Eq. 4) relative to financial self-interest (payoff).

$$U_2 = \pi_2 \tag{2}$$

$$U_2 = \pi_2 - \Theta \cdot \left( \pi_2 / \left( 10 - I + I \cdot M_2 \right) - 1/2 \right)^2 \tag{3}$$

$$U_2 = \pi_2 - \Theta \cdot \left( \left( E_2(E_1(S_2)) - S_2 \right) / \left( E_1(M_1) \cdot I \right) \right)^2 \tag{4}$$

In the guilt aversion model and the MS model, the second-order expectation ($E_2 E_1(S_2)$) is set to half the amount the Investor believes the Trustee has ($E_2(E_1(S_2)) = \frac{1}{2} \times E_1(M_1) \times I$). This is a deviation from prior work[13], where participants' individual second-order beliefs entered into a computational model. However, fixing second-order expectations across participants has the advantage of (a) improving generalizability of the model to new populations which may not have measured second-order beliefs, (b) being consistent with the model simulations in the clustering analysis, where we do not model individual variability in beliefs, and (c) any observed differences in model fit can be explicitly attributed to differences in sensitivity to those beliefs (i.e., guilt aversion) rather than differences in the beliefs themselves. Self-report data confirmed that the expectation model ($E_2(E_1(S_2)) = \frac{1}{2} \times E_1(M_1) \times I$) is an accurate representation of the Trustees' average second-order expectations (see Supplementary Figure 1). Furthermore, Supplementary Figure 11 demonstrates that while using fixed second-order expectations across participants leads to a slight numerical improvement in mean model fit compared with a model with individual second-order expectations, this improvement is not statistically significant across participants.

Across all models, the predicted strategy for the Trustee was the strategy which yielded maximal utility:

$$\hat{S}_2 = \underset{S_2}{\arg\max} \, U_2(S_2) \tag{5}$$

Model performance was measured and compared using the AIC (Eq. 6)[61,62], which rewards model fit and penalizes model complexity (number of free parameters). We chose to use AIC over the alternative Bayesian information criterion, since AIC is superior to BIC if the true data-generating model is not in the model set[63], which is likely true for the current experiment. Assuming that the model errors are normally distributed, AIC is defined as

$$\text{AIC} = n \cdot \ln(\text{SSE}/n) + k \cdot 2 \tag{6}$$

where SSE represents the residual sum of squares (i.e., the sum over squared differences between model prediction and actual behavior), $n$ represents the number of observations (trials), and $k$ represents the number of free parameters in the model (theta and/or phi). We computed AIC per subject, and used one-sample $t$ tests on the subject-wise AIC differences between two models for model comparison[64]. Participants whose behavior was perfectly explained by any model were excluded from model comparisons, since the logarithm of 0 is undefined. We found that average model fit for the Moral Strategy Model was better than all three competing models (Fig. 2c). Two examples of the utility curve described by the Moral Strategy Model in an experimental trial are presented in Supplementary Figure 12.

To ensure that our model was not overfitting the data, and to estimate the stability of decisions within subjects over time, we performed cross-validation on the model predictions. For this step, we divided the 76 trials with nonzero investment for each participant into five equal parts (fivefold cross-validation). For each fold, we fit the model to the remaining 4/5 of the data, and predicted the behavior in the held-out 1/5. We compared the model predictions to the true held-out data across all folds by computing the mean squared prediction error per trial, the Pearson correlation coefficient $r$, and $r^2$. We tested prediction quality with a $t$ test on the $r$ values against 0. In this test, we excluded one participant for whom both the model predictions and the actual behavior were always to return 0 tokens, which caused the correlation to be unidentifiable. Overall, we observed a high degree of model accuracy (mean squared error per trial = 5.37, mean $r^2 = 0.86$; one-sample $t$ test on $r$ values: $t(55) = 66.1$, $p < 0.001$).

We ran parameter recovery analyses to ensure that our model was robustly identifiable. To this end, we created 57 simulated subjects by simulating task data at 57 random points in the theta–phi parameter space, and fit our model to these fake subjects (1000 iterations of the fitting algorithm per simulated subject, best fit selected). The correspondence between the true and recovered parameters was very high (correlation between true and recovered theta: $r = 1.00$, $p < 0.001$; phi: $r = 0.93$, $p < 0.001$). We additionally tested the correspondence by the true simulated task data and the task data predicted based on the recovered parameters; correspondence here was very high as well (mean over 57 fake subjects: mean $r = 1.00$, one-sample $t$ test on $r$ values against zero: $t(55) = 10889.27$, $p < 0.001$).

**Clustering participants by moral strategy**. We aimed to cluster our participants into moral strategy groups without being biased by the particular distribution of moral strategies in our sample. To this end, we first applied hierarchical clustering to simulations of our Moral Strategy Model, and then grouped our participants' behavior by the cluster boundaries obtained from the simulations. Specifically, we created 10,201 (101 × 101) simulated Trustee behavior sets at evenly spaced points in the model's parameter space, with theta ranging from 0 to 0.5 and phi from −0.1 to 0.1, for each combination of investment (0 to 10) and multiplier (2, 4, and 6). We computed the pairwise squared Euclidean distances between these simulations and used the hierarchical clustering algorithm from the *Scipy* package in Python to group the simulations into four parsimonious clusters. Qualitatively, the simulations in these clusters aligned with the theoretical predictions of the four moral strategies we aimed to capture (see Fig. 2b). Finally, we assigned each participant to the cluster of the simulation to which that participant was nearest in parameter space (based on Euclidean distance), thus creating four moral strategy groups.

The result of this approach is a clustering of participants that is not determined by their prevalence in our dataset, or even by their apparent grouping in the model parameter space, but purely based on the theoretical boundaries in the model space. For example, even if our sample had contained no guilt-averse (GA) participants, we would still have identified this strategy in the model simulations, without placing any participants in the corresponding section of the theta–phi parameter space. If, in this case, we had instead used a k-means clustering algorithm set to find four participant clusters, this algorithm would have divided the three participant groups (IA, MO, and GR) into four, thus leading to erroneous clustering of the participants' moral strategies. To display the result of our simulation-based clustering method, and allow readers to judge its reliability, we have plotted all participants' task behavior, grouped by moral strategy, in Supplementary Figure 5A–D.

**Behavioral differences between strategy groups**. There were no significant differences between the groups in gender (chi-square(3) = 2.05, $p = 0.56$) or age ($F(3,53) = 1.65$, $p = 0.19$). There were also no differences in consistency of choices, as measured by the mean squared model prediction error per trial per participant ($F(3,53) = 0.68$, $p = 0.57$). The Guilt Inventory scores (split by state guilt, trait guilt, and moral standards) were also not different between the groups (state guilt: $F(3,53) = 0.37$, $p = 0.77$; moral standards: $F(3,53) = 0.98$, $p = 0.41$; trait guilt: $F(3,53) = 1.23$, $p = 0.31$). The only metric that was different between the four groups was Social Value Orientation (overall effect of group on SVO angle: $F(3,53) = 5.84$, $p = 0.0016$). Post-hoc pairwise comparisons showed that inequity-averse participants had a significantly higher SVO angle (meaning stronger prosocial orientation) than GR and MO participants (IA-GR: $t(29) = 4.45$, $p < 0.001$; IA-MO: $t(43) = 3.43$, $p = 0.0014$); no other pairwise comparisons were significant. This implies that the strategy employed by the inequity-averse participants in our task corresponds to the strategy they employ in the social value orientation task, where higher scores indeed correspond to inequity-averse preferences and lower scores

correspond to greedy preferences[33]. Thus, the behavior of inequity-averse and greedy participants on the HMTG generalizes to the SVO task, which lends construct validity to our task, computational model, and method of clustering participants. Neither guilt aversion nor moral opportunism is explicitly indexed by social value orientation.

The three groups with identical predictions in the ×4 condition (IA, GA, and MO) did not differ on actual choice behavior in this condition. We tested this using linear mixed-effects regression using the *lme4* version 1.1–18–1 and *lmerTest* version 3.0–1 packages in *R* version 3.5.1 (regression model formula: Amount_Returned ~ Investment + Group + (1 + Investment | Subject)). There was no main effect of group on number of tokens returned: $F(2,47) = 2.61$, $p = 0.084$; Supplementary Figure 9). Accordingly, these three groups also did not differ on number of tokens earned in the ×4 condition (one-way analysis of variance (ANOVA) on sum earned in ×4, effect of group: $F(2,47) = 2.51$, $p = 0.092$).

**Replication study**. We conducted a direct replication of this imaging study in a separate behavioral experiment ($n = 102$). This replication sample was part of a larger follow-up study on the Hidden Multiplier Trust Game, which will be described elsewhere. Participants played 80 rounds of the HMTG in the role of Trustee, with a different anonymous Investor each time. The experimental design was identical to the experiment as described in the current paper, with three notable exceptions: (1) the participant was seated in a private behavioral lab space, instead of lying in an MRI scanner; (2) the participant responded to Investor investments using a button-controlled slider that moved in steps of 1 token, instead of in 10% increments (as in the scanner study); (3) the participant played two blocks of 80 trials of this task: one block with multipliers ×2, ×4, and ×6 (as in the imaging study), and one with multipliers ×4, ×6, and ×8 (Investor believes in ×6 multiplier). For the replication analyses in the current paper (Supplementary Figure 4), only the data from the direct replication block (i.e., ×2, ×4, ×6) were used. Block order was counterbalanced across subjects.

**fMRI preprocessing**. Prior to preprocessing, the four read-outs acquired per TR in the multi-echo procedure were realigned and combined per run, using the echo weighting estimated from the first 30 TRs acquired at the start of the run[65]. Motion parameters obtained during realignment were stored and added to the GLM analysis as nuisance regressors. Next, fMRI data preprocessing was carried out using SPM12 (Statistical Parametric Mapping; Wellcome Trust Centre for Neuroimaging, London, UK) in MATLAB version 2014a. Preprocessing of the functional images consisted of slice time correction to the middle slice, coregistration to the T1-weighted anatomical scan, normalization to MNI space (Montreal Neurological Institute) using the deformation fields obtained by segmenting the anatomical scan, and smoothing with a Gaussian kernel of 8-mm full width at half maximum.

**fMRI GLM analysis**. We performed temporal data reduction using a standard first level GLM approach. A GLM was constructed for each participant using boxcar regressors for each task condition. All four screens of the task were taken as conditions, with the trials in the decision and response screens split by multiplier level. A parametric modulator for investment size was added during the investment screen. The two runs were modeled by separate regressors in the same GLM. We thus estimated a GLM for each participant with the following regressors per run:

1. Investor identity screen
2. Investment screen
3. Parametric modulator: investment size
4. Decision screen ×2
5. Decision screen ×4
6. Decision screen ×6
7. Response screen ×2
8. Response screen ×4
9. Response screen ×6
10–15. Realignment parameters

To account for residual variance, the temporal derivative of each condition regressor was added to the model as well as a constant regressor for each entire run. The resulting GLM was convolved with SPM's canonical hemodynamic response function. The model was corrected for temporal autocorrelations using a first-order autoregressive model and a standard high-pass filter (cutoff at 128 s) was used to exclude low-frequency drifts. The Decision screen parameter estimates obtained from the first level GLM were used in all subsequent analyses.

**Inter-subject representational similarity analysis (IS-RSA)**. The inter-subject representational similarity analysis was carried out in Python 2.7.12 using the *NLTools* package version 0.3.6 (http://github.com/ljchang/nltools). We first obtained each participant's mean "Decision screen ×4" activity map by averaging over the corresponding GLM beta maps for the two runs. We then divided these subject-level beta maps into 200 parcels using a whole-brain parcellation based on meta-analytic functional coactivation of the Neurosynth database[30] (parcellation available at http://neurovault.org/images/39711/ and displayed in Supplementary Figure 13). The use of a parcellation scheme has several advantages over the more conventional searchlight approach. First, it is several orders of magnitude less

computationally expensive. Second, the parcels are non-overlapping and contain bilateral regions that reflect functional neuroanatomy, whereas a searchlight approach is limited to local spheres that do not adapt to different areas of the cortex.

Next, we created a dissimilarity matrix for each parcel (the "parcel dissimilarity matrices") using pairwise correlation dissimilarity between each pair of participants. Correlation distance is a useful metric that can accommodate data that is on different scales, which is important when comparing different participants' beta maps. We also created a dissimilarity matrix using the Euclidean distance between each pair of participants in the Moral Strategy Model's parameter space. This "model dissimilarity matrix" captured the dissimilarity between participants in their motivations for reciprocity (moral strategy).

We then computed the correlation between each parcel dissimilarity matrix and the model dissimilarity matrix using Spearman's rank-order correlations on the lower triangle of the matrices. To obtain significance levels of the resulting Spearman's rhos, we computed the same statistic after shuffling the order of the observations in one of the two matrices 10,000 times, and calculated the proportion of instances in which the permuted rho exceeded the true rho. These Monte Carlo *p*-values were Bonferroni-corrected by multiplying them by the number of parcels (200). All *p*-values that remained below 0.05 after this correction were taken to indicate a significant association between model distance and parcel representation distance, and thus a significant relationship between moral strategy and multivariate brain activity patterns in a given parcel.

It is important to note that in this analysis we are not directly comparing brain responses between two groups, and we are not assuming a linear relationship between an individual difference variable and intensity in a single voxel. Such traditional analyses assume Gaussian error and require a large number of participants to generate adequate statistical power to confidently determine that the effect described by the first moment of the distribution is significant given the variance. Instead, we are examining the second moment of a multivariate distribution. This type of analysis is invariant to mean differences in individual activity and explicitly focuses on covariance between participants (see ref. [66]). There are no standard methods to calculate power for this type of analysis, but we are calculating similarity across 1596 pairwise observations ($57 \times 56/2$) for these analyses, as compared with 57 subject-wise observations in a standard analysis. This analysis also allows us to examine the similarities between groups, as opposed to solely examining average differences between groups as per a standard univariate approach.

**Cluster strength analysis**. To evaluate the degree to which a given moral strategy exhibited a unique brain representation, we calculated the cluster strength metric. This metric (Eq. 7) is similar to cluster validity metrics used in unsupervised machine-learning applications[34] and is a normalized metric between $[−1,1]$ that is calculated by subtracting the mean representational dissimilarity (1−correlation) of that participant to the other participants in the same moral strategy group ("within") from the mean dissimilarity to all participants in the other groups ("between") and normalizing by the greatest of the two:

$$\text{cluster strength} = \frac{\overline{\text{between}} - \overline{\text{within}}}{\max\left(\overline{\text{between}}, \overline{\text{within}}\right)} \tag{7}$$

Positive cluster strength values indicate that the participant clusters together with others of the same moral strategy in the parcel's *n* dimensional representational space ($n = $ number of voxels in the parcel) and away from other moral strategies.

To provide an intuition for the cluster strength score, we visualized our approach in Supplementary Figure 14 using a reduced dimensional space. In the middle panel of this figure, we plotted a two-dimensional simplification of the left DLPFC's 745-dimensional space (745 voxels; ×4 condition) based on a multidimensional scaling (MDS) dimensionality reduction of activity patterns in this region. The MDS algorithm was applied using the *Hypertools* package version 0.4.2 for Python after *z*-scoring the patterns within each participant. The 2d projection shows that guilt-averse participants cluster together in the left DLPFC compared to the other participants. Therefore, guilt-averse participants have activity patterns that are more similar to one another than to the participants in the other moral strategy groups. The right panel summarizes this relationship in the form of the cluster strength score, with participants grouped by moral strategy and rank-ordered within each group, as is customary for silhouette plots[35].

We tested whether a brain region was significantly associated with a given moral strategy by permuting the sign of the cluster strength scores for all participants in this strategy group, and then evaluating whether this group's average cluster strength was statistically distinguishable from zero (i.e., meaningful clustering). In this step, we permuted the sign of the observed cluster strength scores within a moral strategy group 5000 times and compared the actual mean cluster strength in the moral strategy group to the distribution of the permuted scores. The Monte Carlo *p*-value was the proportion of permuted scores exceeding the actual score. Although this metric accounts for differences in sample sizes for each moral strategy, it is important to note that smaller groups have less power.

To test whether strategy-specific activity patterns in ×4 condition generalize to ×2 and ×6, we again used the cluster strength score, but now measured across conditions. For example, we tested whether an IA participant's activity pattern in the ×4 condition was more similar to other IA participants' ×2 patterns than to

non-IA participants' ×2 patterns. We again tested the association between a strategy and a brain parcel by permuting the sign of the cluster strength scores of that strategy group in that parcel. The conjunction with the within-×4 clustering test yields the "strategy maps" (Fig. 4f), which highlight regions where strategy groups clustered in ×4, and where the associated activity patterns generalized to ×2 and ×6.

**Similarity analysis of Moral Opportunists to GA and IA**. To determine whether the Moral Opportunists flexibly expressed the multi-voxel activity patterns that we found in the guilt-averse and inequity-averse participants, we computed the similarity of MO participants' activity patterns to the GA–IA pattern difference map in each parcel and each condition. We created these GA–IA difference maps in the following way. We first extracted each GA and IA participant's mean decision screen beta maps for the two conditions from the GLM results, and standardized (z-scored) them per participant, per condition, and per parcel. We then averaged the z-scored maps per group (GA/IA), condition (×2/×6), and parcel, and computed the differences between the two groups' mean z-maps. This yielded, for each parcel and condition, a GA–IA pattern difference map with positive values for voxels that were commonly active in guilt-averse participants relative to inequity-averse participants, and negative values for the reverse. We next calculated the spatial similarity (Pearson correlation) of the MO activity patterns for each condition (i.e., ×2 or ×6) in each parcel to the corresponding GA–IA pattern difference maps.

**Reporting summary**. Further information on experimental design is available in the Nature Research Reporting Summary linked to this article.

## Data and code availability
All data and custom code required to reproduce the results in this paper are available from the Donders Institute for Brain, Cognition, and Behavior repository at: http://hdl.handle.net/11633/aabwlrrn. The analysis code can additionally be found at https://github.com/jeroenvanbaar/MoralStrategiesFMRI.

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

## Acknowledgements

This research was supported by European Research Council project 313454 (to A.S.) and the National Institute of Health R01MH116026, R56MH080716 (to L.C.). We thank Felix Klaassen for his help in data collection.

## Author contributions

A.S. and J.v.B. designed the study; J.v.B. performed the experiments; all authors designed the computational model; J.v.B. and L.C. analyzed the data; all authors wrote the paper.

## Additional information

**Competing interests:** The authors declare no competing interests.

