## [Peer Review File · Nature Communications]

Reviewers' comments:

Reviewer #1 (Remarks to the Author):

van Baar and colleagues present new data to address previously raised concerns of the reviewers. Most of my earlier concerns have been addressed. I have one remaining unaddressed question that was perhaps not phrased clearly enough in my previous review (sorry about that). This concerns the question of whether the observed brain representational differences between strategy groups reflect differences in mental operations happening at the trial level. Another way of phrasing this question is whether the observed differences are specific to the decision phase, which would seem to be necessary to claim that these differences reflect differences in mental operations occurring during decision-making.

To test this, one could repeat the representational similarity analyses performed on data collected during the decision screen, but this time on data collected during non-decision phases of the scanning session (e.g. ITI). If the authors' claims are correct, then the representational differences between the groups should be strongest during the decision phase, compared to other parts of the task. This would be a straightforward analysis to run since it would only involve changing the time bins that are input into the analysis. I would feel much more confident in the authors' claims that representational differences between groups are specific to decision-level computations if they can show that these differences are strongest in the decision phase.

Reviewer #3 (Remarks to the Author):

The authors have adequately responded to my first comment. On the other hand, I'm not persuaded by their response to my other concern. However, I do appreciate the novel methods that the authors have used in this manuscript, and I do think that the findings about the moral phenotypes will likely be of interest to the field. Also, I do think that the manuscript is already very busy. For these reasons, I don't want to hold up the manuscript any further with any additional requests.

Reviewer #1 (Remarks to the Author):

van Baar and colleagues present new data to address previously raised concerns of the reviewers. Most of my earlier concerns have been addressed. I have one remaining unaddressed question that was perhaps not phrased clearly enough in my previous review (sorry about that). This concerns the question of whether the observed brain representational differences between strategy groups reflect differences in mental operations happening at the trial level. Another way of phrasing this question is whether the observed differences are specific to the decision phase, which would seem to be necessary to claim that these differences reflect differences in mental operations occurring during decision-making.

To test this, one could repeat the representational similarity analyses performed on data collected during the decision screen, but this time on data collected during non-decision phases of the scanning session (e.g. ITI). If the authors' claims are correct, then the representational differences between the groups should be strongest during the decision phase, compared to other parts of the task. This would be a straightforward analysis to run since it would only involve changing the time bins that are input into the analysis. I would feel much more confident in the authors' claims that representational differences between groups are specific to decision-level computations if they can show that these differences are strongest in the decision phase.

We appreciate the willingness of Reviewer 1 to clarify their thoughtful concern. We agree that it would be relevant to show that the inter-subject RSA effects, which we believe reflect inter-subject differences in decision-related brain processes, are specific to the decision phase of the Hidden Multiplier Trust Game. To this end, we have carried out the analysis suggested by Reviewer 1, repeating the main representational similarity analysis for data recorded during non-decision phases of the experimental task.

In the figure below (which we include as supplementary figure 7 in our manuscript and reference in the main text on page 12 line 244-245) we show the IS-RSA effects for each of the four screens of the task: the Player, Investment, Decision, and Response screens. These plots highlight the statistical distribution (middle) and anatomical location (right) of brain parcels where a significant positive relationship was found between inter-subject distance in decision strategy (as measured by distance in computational model parameter space) and inter-subject distance in mean parcel activity pattern. As this figure illustrates, these IS-RSA effects are by far the most prevalent during the decision screen, which supports our interpretation of these effects as reflecting decision-related brain processes. The IS-RSA effects taper off in the Investment and Response screens, and all but disappear in the decision-irrelevant Player introduction screen.

Supplementary Figure 7. Comparison of inter-subject representational similarity (IS-RSA) effects between the four phases of the HMTG task—the Player, Investment, Decision, and Response screen. IS-RSA effects were by far the most prevalent in the Decision phase of the task, and virtually non-existent when players were introduced to the investor in the Player screen. Left panels: Example screens. Middle panels: Histogram of IS-RSA effects across all 200 brain parcels. IS-RSA effects expressed as Spearman correlation (Rho) between inter-subject distance in computational model parameter space and inter-subject distance in brain parcel representational space, with statistical cutoff at $P < 0.05$ (Bonferroni-corrected over 200 brain parcels). Right panels: Thresholded brain maps displaying the parcels where a significant positive IS-RSA effect was observed.

REVIEWERS' COMMENTS:

Reviewer #1 (Remarks to the Author):

All of my remaining concerns have been addressed - the new analysis of different task stages is very convincing. I am happy to support publication.